# Quantitative Connection between Cell Size and Growth Rate by Phospholipid Metabolism

**DOI:** 10.3390/cells9020391

**Published:** 2020-02-08

**Authors:** Zhichao Zhang, Qing Zhang, Shaohua Guan, Hualin Shi

**Affiliations:** 1School of Physical Sciences, University of Chinese Academy of Sciences, Beijing 100049, China; zhangzhichao@mail.itp.ac.cn (Z.Z.); guansh@itp.ac.cn (S.G.); 2CAS Key Laboratory of Theoretical Physics, Institute of Theoretical Physics, Chinese Academy of Sciences, Beijing 100190, China; qzhang519@gmail.com

**Keywords:** genome-scale metabolic network, flux balance analysis, cell size, *Escherichia coli*

## Abstract

The processes involved in cell growth are extremely complicated even for a single cell organism such as *Escherichia coli*, while the relationship between growth rate and cell size is simple. We aimed to reveal the systematic link between them from the aspect of the genome-scale metabolic network. Since the growth rate reflects metabolic rates of bacteria and the cell size relates to phospholipid synthesis, a part of bacterial metabolic networks, we calculated the cell length from the cardiolipin synthesis rate, where the cardiolipin synthesis reaction is able to represent the phospholipid metabolism of *Escherichia coli* in the exponential growth phase. Combined with the flux balance analysis, it enables us to predict cell length and to examine the quantitative relationship between cell length and growth rate. By simulating bacteria growing in various nutrient media with the flux balance analysis and calculating the corresponding cell length, we found that the increase of the synthesis rate of phospholipid, the cell width, and the protein fraction in membranes caused the increase of cell length with growth rate. Different tendencies of phospholipid synthesis rate changing with growth rate result in different relationships between cell length and growth rate. The effects of gene deletions on cell size and growth rate are also examined. Knocking out the genes, such as Δ*tktA*, Δ*tktB*, Δ*yqaB*, Δ*pgm*, and Δ*cysQ*, affects growth rate largely while affecting cell length slightly. Results of this method are in good agreement with experiments.

## 1. Introduction

Cell size is an important measurable characteristic in the study of bacterial physiology. Since the growth rate is another significant physiological feature, the link between them is worth studying. A significant phenomenon about bacterial growth is that cell volume increases with the growth rate under nutrient limitation [1,2,3]. This relationship, known as Schaechter’s growth law, was discovered 60 years ago [1]. Experiments studying cell size and growth rate were performed with different strains of *Escherichia coli* under various growth conditions [4,5,6,7,8,9,10,11,12,13]. The Schaechter’s growth law has been reconfirmed at the single-cell level. In addition, two new kinds of relationships have been discovered. One is that cell size is independent of growth rate under translation inhibition [10] and another is cell size decreases with growth rate under LacZ overexpression [10]. Although cell size control and cell growth involve a variety of processes, cell length and growth rate follow a simple relationship. Therefore, the coordination between cell size and growth rate is critical to understand the growth features of bacteria.

Bacterial cells regulate fatty acid synthesis and cell membranes are assembled to form a proper membrane structure [5,14,15]. As the size of cell membrane relating to phospholipid metabolism is sufficient to quantify the cell size, we intended to link cell size to the growth rate from the aspect of the genome-scale metabolic network. Previous works revealed the relationship between them from many aspects [1,5,6,9,11,12,13,14,16,17,18,19,20,21,22,23,23] but did not propose a quantitative model to describe how cell size and growth rate correlate with each other from the aspect of bacterial metabolism. Since the empirical relationship between cell size and growth rate is suitable for different bacterial strains [1,2,3] and mutants [6,9,10,13] whose growth regulations are not exactly the same, the detailed regulation mechanism may be not critical to the correlation between the growth rate and cell size. Regardless of the specific regulations of bacterial growth, the metabolic-balance constraint will establish a relationship between the synthesis rates of materials required for cell membranes and for other parts of cells. It results in the correlation between bacterial cell size and growth rate.

We provided a method to estimate bacterial cell length from the cardiolipin synthesis rate. Considering that phospholipid is a main constituent of the bacterial membranes, we estimated the cell surface area from the phospholipid amount in a cell based on the bacterial envelope structure. The Michaelis–Menten kinetics of the cardiolipin synthesis reaction, which is able to represent all the phospholipid synthesis in the bacterial exponential growth phase, sets another constraint between the amount of phospholipid and the cell surface area. This two constraints set the quantitative relationship between the cell length and the cardiolipin synthesis rate. Combining this method with the relationship between the growth rate and the cardiolipin synthesis rate, we finally obtained a function about the cell length and the growth rate. Two ways were introduced to quantify the relationship between the growth rate and the cardiolipin synthesis rate. One way was the flux balance analysis (FBA) of the genome-scale metabolic network. FBA [24,25,26,27] is a widely used method to predict the growth rate and metabolic reactions rates. It is suitable to analyze metabolic features of a single cell organism growing in the exponential phase and it has been applied to several organisms such as *H. influenzae* [28,29], *S. cerevisiae* [30,31], and *E. coli* [32,33,34,35,36,37,38,39]. FBA provides the relationship among the growth rate and multiple metabolic reactions including the cardiolipin synthesis. Since FBA is also successful in analyzing the effects of gene deletions and drug inhibitions [40,41,42], we analyzed the gene mutants’ effects on cell size and growth rate. Another way was directly using a linear equation to quantify the relationship between the growth rate and the cardiolipin synthesis rate. By changing the linear coefficient of this equation, we had the different relationship between the cell length and the growth rate and predicted that cell length can decrease or be unchanged as growth rate increases.

## 2. Materials and Methods

### 2.1. Cell Surface Area Connected to the Amount of Phosphatidylglycerol

Based on the envelope structure of gram-negative bacteria, the cell surface area of *Escherichia coli* is a function of the amount of phospholipid. The envelope of *Escherichia coli* mainly has three layers: inner membrane, periplasm, and outer membrane [2,43,44,45,46,47,48]. Both inner membrane and outer membrane take a phospholipid bilayer as the skeleton. A layer of lipopolysaccharide covers the outer membrane and a peptidoglycan layer exists in the periplasmic zone. Various enzymes and molecular machines are inserted into or covered on the inner and outer membrane [45,49,50]. As both inner membrane and outer membranes are mainly composed of a phospholipid bilayer with embedded proteins [44,45,50,51,52,53], the surface area of a bacterial cell is divided into two parts. One is the area covered by two phospholipid bilayers, and the other is the area covered by proteins embedded in the membranes. This is written as
(1)$$s_{\mathrm{area}}=s_{\mathrm{lipid}}+s_{\mathrm{protein}}.$$

Here, $s_{\mathrm{area}}$ denotes the surface area of a bacterial cell. $s_{\mathrm{lipid}}$ is the surface area covered by phospholipid in membranes of a cell, and $s_{\mathrm{protein}}$ is the surface area covered by embedded proteins in membranes of a cell, drawn as Figure 1A. Using *f_l_* representing the fraction area of phospholipid in membranes, we had
(2)$$s_{\mathrm{area}}=\frac{s_{\mathrm{lipid}}}{f_{l}}$$

As the area covered by a phospholipid molecule *s_p_* is about 0.5 square nanometers [54], we had
(3)$$s_{\mathrm{lipid}}=\frac{1}{4}\cdot n_{\mathrm{lipid}}\cdot N_{A}\cdot s_{p}.$$

Here, *n*_lipid_ denotes the amount of phospholipid in two layers of membranes in a cell in unit of mol. *N*_A_ is the Avogadro constant in unit of mol^−1^. The factor $\frac{1}{4}$ derives from two parts. One is that each cell of *Escherichia coli* has two layers of membranes, and the other is that each membrane is a lipid bilayer. The cartoon is shown as Figure 1A.

Since cell polar caps are relatively unchanged [55,56] and new cell caps are formed in the middle of the cell during division period [57], the phospholipid in membranes is classified into two parts: one part is the lipid in the cell side surface and the other part is the lipid in cell caps. The idealized shape of *E. coli* used in this work is a right circular cylinder with two hemispherical polar caps, shown as Figure 1A. The cell side surface ($s_{\mathrm{side}}=\pi D\left(L-2\cdot \frac{D}{2}\right)$) refers to the lateral area of the right circular cylinder, shown as the gray area in Figure 1A, and the cell caps ($s_{\mathrm{caps}}=\pi D^{2}$) are the hemispherical polar caps shown as the purple area in Figure 1A. Thus, the amount of phospholipid in membranes ($n_{\mathrm{lipid}}$) is the amount of phospholipid in cell side area ($n_{\mathrm{side}}$) plus the amount of phospholipid in cell caps area ($n_{0}$), written as:(4)$$n_{\mathrm{lipid}}=n_{\mathrm{side}}+n_{0}.$$

Since a bacterial cell has two semisphere caps at the beginning of a cell cycle and has four semisphere caps at the end of a cell cycle [55,56], the average value of $n_{0}$ during a cell cycle is about the phospholipid in ϵ times semisphere caps. ϵ is average number of semispheres during a cell cycle. Thus, we had
(5)$$n_{0}=\frac{f_{l}\cdot \epsilon \frac{\pi D^{2}}{2}}{\frac{s_{p}\cdot N_{A}}{4}}.$$

In order to estimate the amount of all kinds of phospholipid in a bacterial cell from the amount of phosphatidylglycerol (abbreviated as PG), we introduced the factor *f*_PG_ as the ratio of the amount of PG in cell side surface to all phospholipid in cell side surface. We assumed *f*_PG_ was equal to the ratio of the amount of PG to lipid in bacterial cells, which is about 18% [35,58]. The amount of PG in cell side surface is denoted as *n*_PG_. Then we had
(6)$$n_{\mathrm{lipid}}=\frac{n_{\mathrm{PG}}}{f_{\mathrm{PG}}}+n_{0}.$$

Inserting Equations (Equation 6) and (Equation 3) into Equation (Equation 2), we had
(7)$$s_{\mathrm{area}}=\frac{N_{A}\cdot s_{p}}{4\cdot f_{l}}\cdot \left(\frac{n_{\mathrm{PG}}}{f_{\mathrm{PG}}}+n_{0}\right).$$

Because
(8)$$\begin{array}{ccc} s_{\mathrm{area}} & = & s_{\mathrm{side}}+s_{\mathrm{caps}}, \end{array}$$
(9)$$\begin{array}{cc} = & \pi D\left(L-2\cdot \frac{D}{2}\right)+\pi D^{2}, \end{array}$$
(10)$$\begin{array}{cc} = & \pi DL, \end{array}$$

Equation (Equation 7) was rewritten as
(11)$$\pi DL=\frac{N_{A}\cdot s_{p}}{4\cdot f_{l}}\cdot \left(\frac{n_{\mathrm{PG}}}{f_{\mathrm{PG}}}+n_{0}\right),$$
an equation about cell length (*L*), cell width (*D*), and the amount of PG (*n*_PG_).

### 2.2. Estimation of Phosphatidylglycerol Amount from the Kinetics of the Cardiolipin Biosynthesis Reaction

Based on the kinetics of an enzyme-catalyzed reaction, the reaction rate is a function of the reactant concentration and the enzyme concentration. To obtain the PG concentration, we selected the cardiolipin biosynthesis reaction catalyzed by the cardiolipin synthase clsA [59,60,61,62,63], because the products of the cardiolipin biosynthesis reaction [61,64] are also a kind of major phospholipid in cell membranes. During a cell cycle, cardiolipin must be produced, and *Escherichia coli* in the exponential growth phase synthesizes the cardiolipin due to the activity of clsA [65,66,67,68]. The cardiolipin biosynthesis reaction occurs in the inner membrane and the phospholipid of the outer membrane is transported from the inner membrane [45,49,50,52]. According to the Michaelis–Menten kinetics, this reaction is written as
(12)$$2\;\mathrm{PG}+\mathrm{clsA}-\mathrm{free}\overset{k_{1}}{\underset{k_{-1}}{⇌}}\mathrm{clsA}-2\mathrm{PG}\overset{k_{2}}{\to }\mathrm{cardiolipin}+\mathrm{glycerol}+\mathrm{clsA}-\mathrm{free},$$
where $k_{1}$, $k_{-1}$, and $k_{2}$ are the rate constants. This reaction is abbreviated as CLPNS in this work.

According to the law of mass action, the reactions rates are written as
(13)$$\begin{array}{c} r_{1}=k_{1}{\left[\mathrm{PG}\right]}^{2}\left[\mathrm{clsA}-\mathrm{free}\right], \end{array}$$
(14)$$\begin{array}{c} r_{-1}=k_{-1}\left[\mathrm{clsA}-2\mathrm{PG}\right], \end{array}$$
(15)$$\begin{array}{c} r_{\mathrm{CLPNS}}=k_{2}\left[\mathrm{clsA}-2\mathrm{PG}\right], \end{array}$$
where [X], the concentration of X, is the amount of X per area in unit of $\mathrm{mol}/\mu m^{2}$. This is beacause CLPNS takes place at the inner membrane [45,49,50,52]. X stands for any substances such as PG and clsA. Because $\frac{d[\mathrm{clsA}-\mathrm{free}]}{dt}=-r_{1}+r_{-1}+r_{2}$, when the concentration of clsA did not change, we had
(16)$$k_{1}{\left[\mathrm{PG}\right]}^{2}\left[\mathrm{clsA}-\mathrm{free}\right]=k_{-1}\left[\mathrm{clsA}-2\mathrm{PG}\right]+k_{2}\left[\mathrm{clsA}-2\mathrm{PG}\right].$$

Inserting Equations (Equation 13)–(15) into Equation (Equation 16), we had
(17)$$r_{\mathrm{CLPNS}}=V_{\max}\frac{{\left[\mathrm{PG}\right]}^{2}}{K_{M}+{\left[\mathrm{PG}\right]}^{2}},$$
where *r*_CLPNS_ is the reaction rate of CLPNS, PG is the concentration of PG, and *V*_max_ and *K*_M_ are two kinetic parameters. Because the unit of *r*_CLPNS_ in Equation (15) is $\mathrm{mol}/(\mu m^{2}\cdot h)$, a unit transformation was done to have the unit consistent with that in FBA where the unit of *r*_CLPNS_ is mmol/(gDW·h) (Detail information is written in Section A.3). *K*_M_ is about $2\times 10^{-35}{\left(\mathrm{mol}/\mu m^{2}\right)}^{2}$ [62] and *V*_max_ is about 0.38 mmol/(gDW·h) in BW25113 [62,69].

Inserting $\left[\mathrm{PG}\right]=\frac{n_{\mathrm{PG}}}{s_{\mathrm{side}}}$ to Equation (Equation 17), we had
(18)$$\frac{n_{\mathrm{PG}}}{s_{\mathrm{side}}}={\left(\frac{K_{M}\cdot r_{\mathrm{CLPNS}}}{V_{\max}-r_{\mathrm{CLPNS}}}\right)}^{\frac{1}{2}},$$
where *n*_PG_ is the amount of clsA in cell side surface of a cell and *s*_side_ is the side surface area of a cell. As $s_{\mathrm{side}}=\pi D\left(L-2\cdot \frac{D}{2}\right)$, Equation (Equation 18) was rewritten as
(19)$$\frac{n_{\mathrm{PG}}}{\pi DL-\pi D^{2}}={\left(\frac{K_{M}\cdot r_{\mathrm{CLPNS}}}{V_{\max}-r_{\mathrm{CLPNS}}}\right)}^{\frac{1}{2}},$$
an equation about *L*, *D*, *n*_PG_ and *r*_CLPNS_.

From Equation (Equation 5), Equation (Equation 11) and Equation (Equation 19), we had an equation about *L*, *n*_0_, and *r*_CLPNS_, written as
(20)$$L=\left(\frac{\frac{\epsilon }{2}-1}{1-\frac{s_{p}\cdot N_{A}}{4\cdot f_{\mathrm{PG}}\cdot f_{l}}\cdot {\left(\frac{K_{M}\cdot r_{\mathrm{CLPNS}}}{V_{\max}-r_{\mathrm{CLPNS}}}\right)}^{\frac{1}{2}}}+1\right)\cdot {\left(\frac{s_{p}\cdot N_{A}\cdot n_{0}}{2\epsilon \pi \cdot f_{l}}\right)}^{\frac{1}{2}}.$$

Then, we can estimate the cell length (*L*) from the cardiolipin synthesis rate (*r*_CLPNS_).

### 2.3. Introducing the Relationship between Growth Rate and Cardiolipin Synthesis Rate

#### 2.3.1. Cell Length Estimation Based on FBA

As flux balance analysis is efficient to simulate the metabolic rates and the growth rate of bacteria in the exponential growth phase (when metabolic reactions was assumed to be at the steady state, and the detailed description of the system was in Section A.1), we integrated flux balance analysis with Equation (Equation 20) to obtain the relationship between cell length and growth rate. FBA [1,32,35] is represented as
(21)$$\begin{array}{cc} \max: & r_{biomass}, \end{array}$$
(22)$$\begin{array}{cc} \mathrm{subject}\;\mathrm{to}: & S\cdot r=0, \end{array}$$
(23)$$\begin{array}{c} b_{l}<r<b_{u}. \end{array}$$
r is a vector of reactions rates in unit of mmol/(gDW·h). *r_biomass_* equals to the growth rate. S is a stoichiometric matrix, whose element *S_ij_* is the stoichiometry of the metabolite *i* in the reaction *j* in the metabolic network of the system. $b_{l}$ is the lower bounds vector, and $b_{u}$ is the upper bounds vector. Each set of bounds of exchange reactions corresponds to a nutrient medium. By setting different bounds of exchange reactions, we simulated the growth rate and metabolic reactions’ rates including *r*_CLPNS_ of *Escherichia coli* growing in different nutrient conditions. Detail description of FBA was written in Section A.2. Combining FBA with Equation (Equation 20), we are able to examine the effects of nutrient limitation and gene deletions on cell length and growth rate. We called this method the size estimation based on flux balance analysis (SEFBA).

Previous study [70] found that the fraction of protein in cell membranes increases with growth rate and finally reaches to a maximum value. The transport reactions of cells are classified into two groups which are the facilitated diffusion (*r*_fac_) and the simple diffusion (*r*_sim_). The ratio of facilitated diffusion rate (*r*_fac_) to all the transport reactions’ rates (*r*_fac_+*r*_sim_) follows a similar tendency with the fraction of protein in membranes changing with growth rate (shown as Figure A8). In order to introduce the factor that *f_l_* changes with growth rate, we assumed that the ratio of the surface area covered by protein to the surface area covered by phospholipid in membranes was in proportion to the ratio of facilitated diffusion rate (*r*_fac_) to simple diffusion rate (*r*_sim_), written as
(24)$$\frac{s_{\mathrm{protein}}}{s_{\mathrm{lipid}}}=\beta \cdot \frac{r_{\mathrm{fac}}}{r_{\mathrm{sim}}}.$$

This assumption was made because the facilitated diffusion incorporates with membrane proteins to transport substances while simple diffusion depends on the concentration gradient and the membrane area covered by phospholipid. Only small molecules like carbon dioxide, oxygen, and water transport by simple diffusion [54]. The concentrations of oxygen and carbon dioxide in chemostat culture do not change in this work, so the simple diffusion rate is in proportion to the membrane area covered by phospholipid. As for facilitated diffusion, Michaelis–Menten kinetics is able to describe it [71]. When substance concentration is at high value, the facilitated diffusion mainly depends on protein amount in membrane. In other words, the facilitated diffusion rate is approximately in proportion to the membrane area covered by protein at high substance concentration. Thus, we had the Equation (Equation 24) where β is a coefficient. Under this assumption, we had
(25)$$f_{l}=\frac{1}{1+\beta \cdot \frac{r_{\mathrm{fac}}}{r_{\mathrm{sim}}}}.$$
*r*_fac_ and *r*_sim_ were obtained from the results of FBA. The mass fraction of protein to phospholipid in membranes has a maximum value to maintain the structure of membranes, which is about 2.5 (wt/wt) [54,72]. It implies that *f_l_* is larger than 4/9. Because 97% values of $\frac{r_{\mathrm{fac}}}{r_{\mathrm{sim}}}$ obtained from the simulations of iJR904 is smaller than 2.5, the parameter β is estimated to be about 0.5 for iJR904 [35]. Similarly, β is about 0.22 for iJO1366 [32]. When the cardiolipin synthesis rate is small enough, cell length changes slightly with *f_l_*. The influence of *f_l_* on cell size is enhanced with increased synthesis rate of cardiolipin (shown as Figure A9).

To summarize, the final representation of SEFBA is written as:(26)$$\begin{array}{cc} \max: & r_{biomass}, \end{array}$$(27)$$\begin{array}{cc} \mathrm{subject}\;\mathrm{to}: & S\cdot r=0, \end{array}$$(28)$$\begin{array}{c} b_{l}<r<b_{u}, \end{array}$$
(29)$$L=\left(\frac{\frac{\epsilon }{2}-1}{1-\frac{s_{p}\cdot N_{A}}{4\cdot f_{\mathrm{PG}}\cdot f_{l}}\cdot {\left(\frac{K_{M}\cdot r_{\mathrm{CLPNS}}}{V_{\max}-r_{\mathrm{CLPNS}}}\right)}^{\frac{1}{2}}}+1\right)\cdot {\left(\frac{s_{p}\cdot N_{A}\cdot n_{0}}{2\epsilon \pi \cdot f_{l}}\right)}^{\frac{1}{2}}.$$

We used two genome-scale metabolic models iJR904 [35] and iJO1366 [32] in order to examine the effects of different metabolic networks on the results. As Figure 2 and Figure A7 show, the results of iJR904 and iJO1366 are similar. The parameters are listed in Table 1.

#### 2.3.2. The Reduced Model of SEFBA

To obtain an explicit relationship between growth rate and cell length, we directly introduced the relationship between the cardiolipin biosynthesis rate and the growth rate with a linear equation instead of the results of FBA. The cardiolipin biosynthesis rate is assumed to change linearly with the growth rate, written as
(30)$$r_{\mathrm{CLPNS}}=w_{\mathrm{CLPNS}}\lambda +v_{0}.$$

Here, *w*_CLPNS_ and *v_0_* are coefficients in this linear equation. *w*_CLPNS_ represents the correlation between $r_{\mathrm{CLPNS}}$ and growth rate λ. When the reactions’ bound vectors, $b_{l}$ and $b_{u}$, are in a certain range, $w_{\mathrm{CLPNS}}$ and $v_{0}$ are constant. Furthermore, when cells grow with the same bottleneck processes, the values of $w_{\mathrm{CLPNS}}$ and $v_{0}$ are constant. In other words, the relationship between $r_{\mathrm{CLPNS}}$ and growth rate λ does not change when bacteria grow with the same limitations. This equation is suitable among a range of growth rates. Theoretically, *w*_CLPNS_ can be negative while *v_0_* is positive during a certain range of growth rates.

Combining Equations (Equation 30) and (Equation 20), we have
(31)$$L=(\frac{\left(\frac{\epsilon }{2}-1\right)}{1-C\cdot {\left(\frac{1}{H-E\cdot \lambda }-1\right)}^{\frac{1}{2}}}+1)D.$$

Here, $C=\frac{s_{p}\cdot N_{A}}{4\cdot f_{\mathrm{PG}}\cdot f_{l}}\cdot K_{M}^{\frac{1}{2}}$, which is derived from the reaction rate constant *K*_M_. Previous study [70] showed that the protein occupancy in membrane almost does not change when growth rate is faster than 0.3 h^−1^. It is reasonable to assume *f*_l_ constant. Then, *C* is a constant parameter. $H=1-\frac{v_{0}}{V_{\max}}$, and $E=\frac{w_{\mathrm{CLPNS}}}{V_{\max}}$. Both *H* and *E* are related to coefficients *w*_CLPNS_ and *v*_0_. *D* stands for the cell width. This equation describes the quantitative relationship among cell length, cell width, phospholipid synthesis, and phospholipid occupancy in membranes. All the parameters used in this work are summarized in Table 1.

## 3. Results

### 3.1. Extracting Cell Size from the Metabolic Mode

Cell size is associated with the structure of the cell envelope. As for gram-negative bacteria such as *Escherichia coli*, the envelope has two layers of cell membranes. Because the skeleton of cell membranes is phospholipid, cell size depends on the amount of phospholipid. As phosphatidylglycerol takes about 18% of the entire phospholipid in *Escherichia coli* [35,58], it sets the geometric constraint among the phospholipid amount, cell length, and cell width (shown as Figure 1B and Section 2.1). The amount of phosphatidylglycerol is also related to phosphatidylglycerol metabolism. The kinetics of the catalytic reaction of cardiolipin biosynthesis determine the relationship between the reaction rate and the phosphatidylglycerol concentration, thus setting another constraint among the amount of phospholipid, cardiolipin synthesis rate, cell length, and cell width (shown as Figure 1B and Section 2.2). Combining these two constraints, we obtained a function of the cardiolipin synthesis rate, cell width, and cell length (shown as Figure 1B and Equation (Equation 20)). Since cardiolipin biosynthesis is part of the metabolic network, the metabolic balance and bacterial growth strategy relate cardiolipin synthesis rate to the whole metabolic network. Based on the flux balance analysis, the genome-scale metabolic model gives the relationships between growth rate and metabolic reaction rates (shown as Section 2.3.1). Integrating the function of cell length and cardiolipin biosynthesis with the relationship between phospholipid metabolism and growth rate, we quantitatively connected cell length with growth rate by metabolic network.

This method enables us to calculate cell length based on cardiolipin synthesis kinetics and the genome-scale metabolic networks, which is different from previous research. Previous models about metabolic networks and flux balance analysis studied the distribution of metabolic reactions and the effects of growth strategies [75], solvent capacity [76], cell volume constraints [27], and cell surface area constraints [39] on global metabolic reactions. Although the influence of phospholipid metabolism on cell size was studied by experiments [14], a quantitative model has not illustrated it.

### 3.2. Coordination between the Growth Rate and Cell Length

Different nutrient concentrations in culture media lead to different nutrient transport rates (Equation (Equation 2) in research [77], shown as Figure 1C). Nutrient transport reactions, classified into simple diffusion and facilitated diffusion, are the processes in which nutrients in the culture medium transport in or out of cells. When the system is at a steady state, nutrient transport rates are equal to nutrient exchange rates (Section A.1 Description of the system). Nutrient exchange reactions are the processes by which nutrients are added in or expelled from the chemostat. In a chemostat culture system, the nutrient exchange rate is determined by the nutrient concentration in culture media, which controls metabolic reaction rates and the growth rate. Thus, we control the nutrient exchange rates to simulate bacterial metabolic features in various culture media. For example, we set the bounds of nutrient exchange rates to correspond to the M9 minimal medium (a kind of culture medium, same to [73]) with glucose as the single carbon source, then changed the glucose exchange rate from 0 to 30 mmol/gDW/h. The results of FBA show that growth rate increases with glucose exchange rate (shown as Figure 1D) and that cardiolipin biosynthesis rate increases linearly with the growth rate (shown as Figure 1E). Equation (Equation 20) depicts the relationship between cardiolipin synthesis rate and cell length (shown as Figure 1F). Finally, cell length is a function of the growth rate. SEFBA demonstrates that lipid biosynthesis links cell size with growth rate (Figure 1D–F), which describes how cell length varies with the growth rate under multiple nutrient limitations. To validate the SEFBA model, we compared it with experimental data [6,9,13] of four bacterial strains. The bounds of exchange reactions in FBA correspond to the availability of nutrients in growth media [35]. Each series of bounds corresponds to a group of growth culture media which had the same nutrient components. We used 20 series of bounds (written in Appendix A) to simulate the metabolic states of *Escherichia coli* in different culture media [6,25,73,74,78]. Altering the maximum rates of bottleneck carbon-source exchange reactions from 0 to 30 mmol/(gDW·h), we obtained 20 series of SEFBA results. Figure 2 compares the results of SEFBA with experimental data.

The solid lines in Figure 2 with different colors correspond to 20 culture media. They are not exactly the same but have a similar tendency, which is consistent with the empirical relationship that the cell size is larger at faster growth rate. Furthermore, it indicates that the tendency of cell length with growth rate does not strongly depend on culture media.

The bias of SEFBA may come from two main reasons. One reason is that the constraints of FBA are off the real condition. For example, neglecting the complicated genetic regulation mechanism results in the losses of coordination among reactions. Another reason is the thermodynamic constraints of reactions [79,80]. The metabolic reactions network has reaction loops [32,35]. This may lead to the large reaction rates in the results of FBA.

### 3.3. Influence of Phospholipid Synthesis and Cell Width on Cell Length

Cell length can be classified into two parts according to membrane formation. One part is the length occupied by cell caps which are formed in the middle of the cell, and another part is the length of cell side surface which extends by adding phospholipid and protein into cell membranes [44,81]. Cell width depends on *n*_0_ the average amount of phospholipid in cell caps during a cell cycle and *f_l_* the fraction area of phospholipid in cell membranes. The length of the cell side surface relates to *f_l_* and cardiolipin biosynthesis rate $r_{\mathrm{CLPNS}}$.

Assuming that *n*_0_ is constant while *f_l_* varies, we had the solid lines in Figure 2. The values of *n*_0_ for each of the bacterial strains were estimated from the average value of cell width. The segments and fluctuations in the solid lines in Figure 2 were derived from the alteration of active metabolic reactions which resulted in the variations of the relationship between *f_l_* and λ and the relationship between *r*_CLPNS_ and λ. As Figure A3 shows, *r*_CLPNS_ is almost in proportion to λ. Thus, the fluctuations of cell length mainly come from the variation of *f_l_* (Figure A3).

Assuming *f_l_* is constant while *n*_0_ varies, we had the black dashed lines in Figure 2. Under nutrient limitation, *v*_0_ equals to zero, which means *r*_CLPNS_ is 0 when λ is infinitely close to 0. *w*_CLPNS_ is the slope of *r*_CLPNS_ changing with λ. Since the light blue area in Figure 2 stands for the cell length variation caused by the cell width change and the black dashed line represents the cell length variation caused by the change of cell width and *r*_CLPNS_, the gray area represents cell length variation derived from phospholipid synthesis. It implies that the influence of phospholipid synthesis on cell length is a little stronger than that of cell width variation in some cases.

To further validate cell width variation with growth rate, we used Equation (Equation 5) to fit the experimental data of cell width for each bacterial strain, shown as Figure A4. In Figure A5, the black lines are the cell length calculated under the condition that *f_l_* is changing while *n*_0_ is constant. The orange lines are the cell length calculated with *f_l_* constant and *n*_0_ changing. Black lines are close to orange lines. The change of either *f_l_* or *n*_0_ leads to the change of cell width. This means that cell width does affect cell length, but the source of variation in cell width is not significant.

### 3.4. Three Forms of Relationship between Cell Length and Growth Rate under Different Stress Conditions

The relationship between cardiolipin synthesis rate and growth rate, obtained by FBA, is not explicit and applies only to the nutrient limitation conditions. To focus on the effects of phospholipid synthesis on growth rate and cell size, we used a linear equation straightforwardly to depict how cardiolipin synthesis rate changes with growth rate and then had the reduced model of SEFBA (Section 2.3.2). Thus, the tendency of cardiolipin synthesis rate varying with growth rate determines how cell length changes with growth rate (Figure 3A). The reduced model of SEFBA was able to study the relationship between cell size and growth rate under various stresses (Figure 3B). The different relationship between *r*_CLPNS_ and growth rate results in the alteration of the relationship between cell length and growth rate. As Figure 3A shows, stresses in *Escherichia coli* can be classified into three groups according to the tendency that cell length changes with the growth rate.

With cell width unchanged, cell length will not change with the growth rate if *w*_CLPNS_ is zero. If a factor affects growth rate but not division process and phospholipid synthesis rate, cell length will be independent of the growth rate under the regulation of this factor. Previous research [10] shows that when translation processes were inhibited by different amounts of chloramphenicol, bacterial size was almost constant while the growth rate changed. This implies that cardiolipin synthesis rate *r*_CLPNS_ is independent of the growth rate under translation inhibition.

Another classification is that cell length decreases with the growth rate. It occurs when *w*_CLPNS_ is negative, which happens when a factor inhibits cell growth but accelerates phospholipid synthesis. Previous studies [10] showed that LacZ overexpression results in reduced growth rate and increased cell length. It implies that *r*_CLPNS_ negatively correlated with the growth rate with lacZ overexpression. Under lacZ overexpression stress, protein allocation strategy is different from that under nutrient limitation, which can cause different metabolic reaction distribution. When LacZ is overexpressed, bacteria may be forced to express more protein, so more phospholipid is needed to enlarge cell volume without lysis. This leads to negative correlation between *r*_CLPNS_ with growth rate.

The last classification is that cell size increases with the growth rate. One typical example is the nutrient limitation. *r*_CLPNS_ positively correlates with growth rate.

### 3.5. Effects of Metabolic Defects on Growth Rate and Cell Length

Impacts of several metabolic gene deletions on cell length and growth rate were researched [13], however, we still lacked a model to predict these effects at a systematic level. With the ability of the genome-scale metabolic model to study gene deletions, we performed flux balance analysis of the metabolic network with genetic defects and calculated cell length to investigate whether cell length showed a positive correlation with growth rate. We did three series of simulations of metabolic gene deletion mutants. Under the influence of metabolic gene deletions, cell length variation correlates to growth rate variation positively (Figure 4A). Knocking out the genes that participate in the phosphate metabolism reduces the growth rate while weakly influencing cell size, such as Δ*tktA*, Δ*tktB*, Δ*yqaB*, Δ*pgm*, and Δ*cysQ*. Some mutants without essential genes cannot grow. These mutants are out of consideration. Furthermore, we examined how the oxygen metabolic pathway affects growth rate and cell size. Interestingly, cell length in anaerobic conditions is larger than that in aerobic conditions, while growth rate in anaerobic conditions is smaller than that in aerobic conditions (see Figure A6).

To further examine the SEFBA model, we compared the SEFBA predictions of gene-deletion strains with experiments [4] which studied the influence of several central carbon metabolism genes on cell size and the growth rate. The medium [4] used to culture these nine mutants was Luria-Bertani (LB) broth supplemented with 0.2% glucose. Since the chemical components of LB are unclear, we did SEFBA with 20 kinds of culture media. The bounds of exchange reactions corresponding to these 20 kinds of culture media were written in Appendix A. For each kind of reaction bound, we did flux balance analysis with various glucose exchange rates and selected the result whose growth rate was closest to the the experimental value. Then, we predicted the cell length of each mutant in these 20 kinds of culture media. The comparison of SEFBA and experiments was drawn in Figure 4B. The large error bars imply that physiology features of mutants depend on the cultural media. It is reasonable that the metabolic defects caused by a gene deletion may be critical in a cultural medium but less critical in another cultural medium. Thus, the cell size of a mutant changes widely among cultural media. The change directions we predicted for Δgnd and ΔacnB are not consistent with experiments. The inconsistency may result from several aspects. One is that components of the FBA model used in this work are not fully constrained, such as the solvent capacity constraint [76], protein allocation constraint [26], etc. Another is that the nutrient uptake rates in the Westfall’s experiments [4] are unavailable—the use of the estimated values of nutrient uptake rates instead of experimental values may lead to mispredictions of some mutants.

## 4. Discussion

This work provides a possible explanation as to why cell size changes differently with growth rate under different kinds of growth stresses from the metabolic aspect. A reason is that the phospholipid metabolic rate changes differently with growth rate under various growth stresses. The uncertainty of parameters in SEFBA among bacterial strains under different growth stresses limits the prediction ability of the model, as well as the reduced model of SEFBA. Despite this, the correlation between growth rate and phospholipid metabolic rate is the key in determining the tendency of cell size changing with growth rate. SEFBA or the reduced model of SEFBA is still able to describe the relationship between cell size and growth rate without considering the detailed regulation processes in cells. It implies the metabolic features are the basis of the correlation between cell size and growth rate.

Unlike previous improved models of FBA [27,39,75,76], our method combined cardiolipin synthesis kinetics with the genome-scale metabolic model to calculate cell length, instead of improving the objective function or the constraints in the flux balance analysis. Thus, this method does not improve the accuracy of metabolic reactions predictions but enables us to predict cell size in various culture media and examine the effects of metabolic modes on cell size. This method accompanied by the conventional FBA is named as SEFBA in this work. It is able to combine with any other improved FBA like FBAwMC [27] and FBA^ME^ [39] instead of the conventional FBA. The accuracy of SEFBA may be improved by replacing the FBA part in SEFBA with other improved FBA models. Besides, if more constraints of reactions could be introduced into the metabolic model, we will be able to study the relationships between bacterial cell size and growth rate with other growth limitations.

The cardiolipin biosynthesis reaction used in this work is the reaction that two molecules of phosphatidylglycerol are transferred into a cardiolipin molecule and a glycerol molecule catalyzed by clsA synthase. Both phosphatidylglycerol and cardiolipin are major phospholipids in bacteria [61,64]. Since the mechanism of cardiolipin biosynthesis reaction [60,63] is much simpler than other phospholipid biosynthesis (for example, PG synthesis processes include two kinds of reactions and involve three enzymes which are pgA, pgB, and MsbA [63]), we used cardiolipin biosynthesis reaction to estimate the concentration of phosphatidylglycerol. As the reaction occurs in exponential growth phase and this method also intends to study the cell size feature of *Escherichia coli* in exponential growth phase, it is suitable to use the cardiolipin biosynthesis reaction to calculate cell size.

SEFBA is not susceptible to the genome-scale metabolic network models. Similar results were obtained with iJR904 and iJO1366 (Figure 2 and Figure A7). Because the transport reactions involved in different metabolic networks are not the same (which results in different simple diffusion rate and facilitated diffusion rate), β needs an update to meet the constraint of phospholipid-to-protein ratio in membranes. The other parameters in SEFBA, including *K*_M_, *V*_max_, and *n*_0_, are independent of metabolic networks.

## Figures and Tables

**Figure 1 cells-09-00391-f001:**
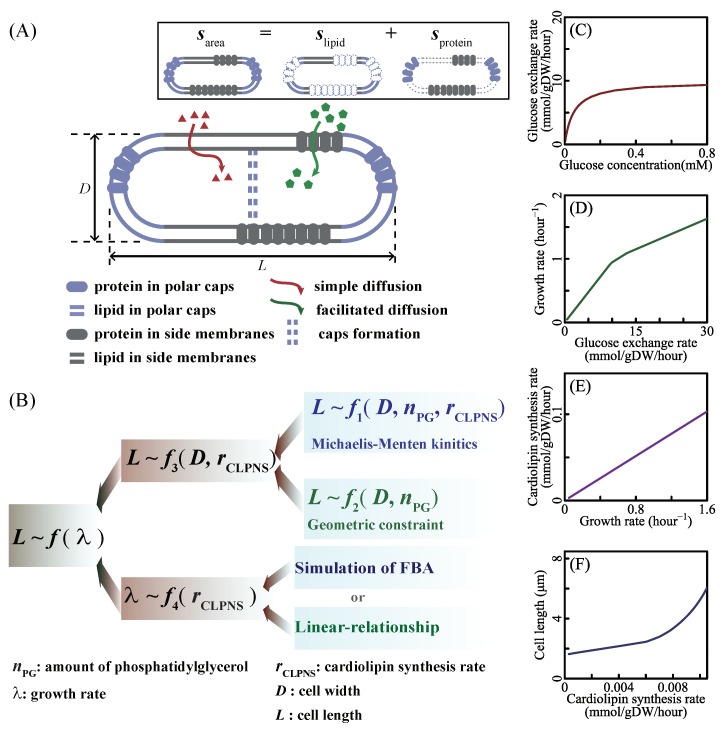
Overview of modeling approaches. (**A**) Gram-negative bacteria have an inner membrane and an outer membrane, both of which are lipid bilayers embedded with membrane proteins. According to the membrane structure, the cell surface area is the function of the amount of phospholipid. (**B**) *f*_1_ refers to the Michaelis–Menten equation of cardiolipin biosynthesis reaction, which sets a constraint among the amount of phosphatidylglycerol, cardiolipin synthesis rate, cell length and cell width (see Equation (Equation 18)). *f*_2_ is another constraint deriving from cell membrane structure (see Equation (Equation 7)). According to *f*_1_ and *f*_2_, *f*_3_ is written as Equation (Equation 20). The relationship between the growth rate and cardiolipin synthesis rate was introduced by two ways in this work. One was based on the flux balance analysis, and another was assuming a linear equation directly. The function of cell length and growth rate was obtained by integrated *f*_3_ and *f*_4_. (**C**–**F**) Simulations of iJR904 [35] whose bounds correspond to the M9 minimal medium [25,73,74] with glucose (written in Appendix A). Exchange rates of nutrients depend on the nutrient concentration. The increase of nutrient exchange rates leads to fast growth rate. The flux balance analysis of genome-scale metabolic network infers that the cardiolipin synthesis rate changes linearly with the growth rate when bacteria grow with same bottleneck reactions. According to Equation (Equation 20), cell length increases with cardiolipin synthesis rate.

**Figure 2 cells-09-00391-f002:**
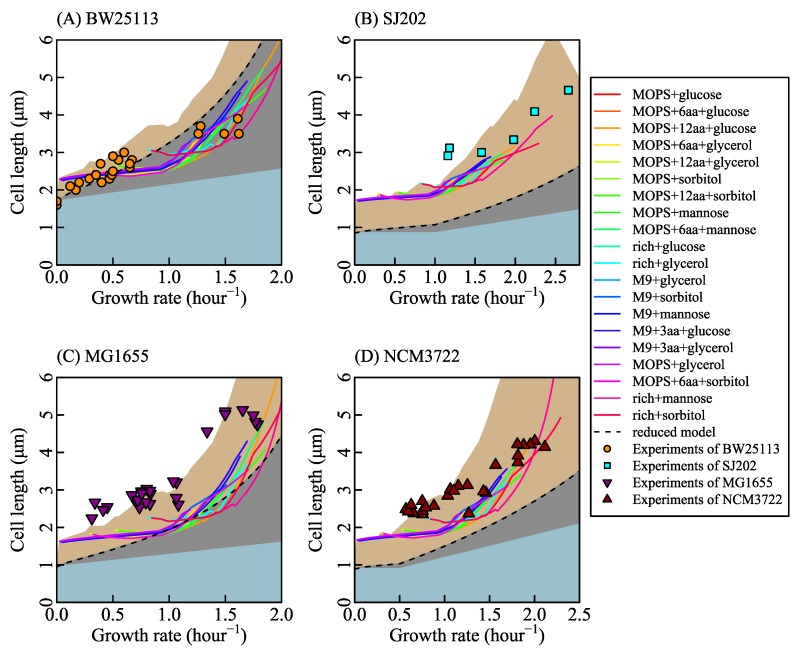
Comparison between the results of size estimation based on flux balance analysis (SEFBA) with experimental data. The solid lines in different colors are the results of SEFBA. Each color of the solid lines stands for a kind of culture medium which is determined by reactions bounds. *K*_M_ is $2\times 10^{-35}{\left(\mathrm{mol}/\mu m^{2}\right)}^{2}$, and β is 0.5 in all sub-figures. The dashed lines were drawn by Equation (Equation 31), where parameter *C* is 2.96 and parameter *H* is 1. Cell width *D* changes linearly with the growth rate based on experiments data [6,9,13]. The parameter *E* is 0.04 in (**A**), 0.023 in (**B**), 0.04 in (**C**), and 0.035 in (**D**), standing for the variation rate of cardiolipin synthesis with growth rate. The upper edges of light blue area were calculated by setting *w*_CLPNS_ zero while changing cell width *D*, so the light blue area represents the cell length variation caused by cell width. The upper edges of light gray area were calculated by setting *w*_CLPNS_ a positive value while changing cell width *D*, so the light gray area represents the cell length variation caused by *r*_CLPNS_. The upper edges of brown area were calculated by setting *n*_0_ changing with growth rate (same to cell width changing with growth rate) in SEFBA. (**A**) The circles in orange are the experimental data [13] with the strain of BW25113. *n*_0_ is $1\times 10^{-16}$. *V*_max_ is 0.38 mmol/(gDW·h). (**B**) The squares in cyan are the measured data from experiment [9] with the strain SJ202. *n*_0_ is $5.8\times 10^{-17}$. *V*_max_ is 0.76 mmol/(gDW·h). (**C**) The inverted triangles filled with purple are from experiments of MG1655 [6]. *n*_0_ is $5\times 10^{-17}$. *V*_max_ is 0.3 mmol/(gDW·h). (**D**) The regular triangles filled with red are experiments of NCM3722 [6]. *n*_0_ is $5.3\times 10^{-17}$. *V*_max_ is 0.38 mmol/(gDW·h). The values of *V*_max_ for BW2513 and MG1655 are based on the mass spectrometric data [69], and the values of *V*_max_ for SJ202 and NCM3722 were obtained by fitting experimental data. The values of *n*_0_ for each of the bacterial strains were estimated based on the cell width. ϵ is 3.4 obtained by fitting the experiments.

**Figure 3 cells-09-00391-f003:**
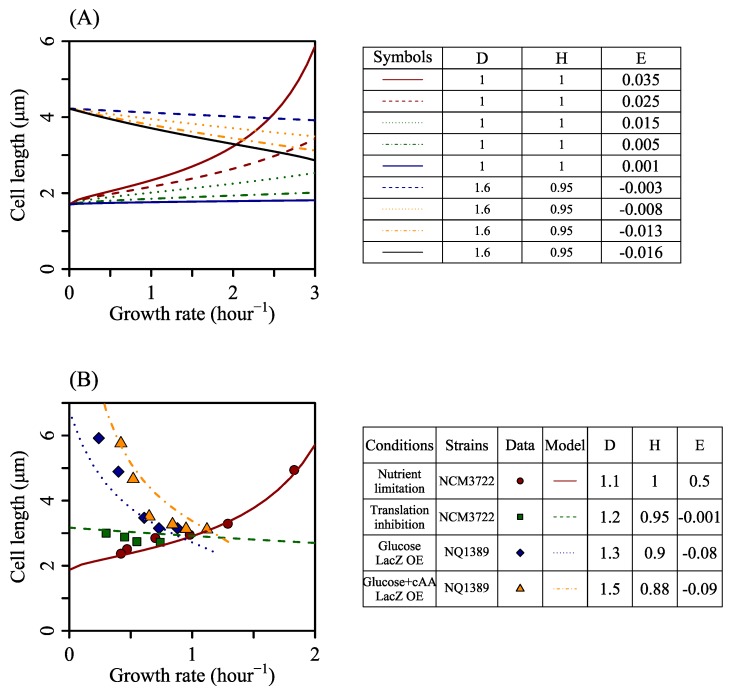
Three tendencies of cell length changing with growth rate. *D* is cell width. $H=1-\frac{v_{0}}{V_{\max}}$, and $E=\frac{w_{\mathrm{CLPNS}}}{V_{\max}}$ (See Equation (Equation 31)). *E* represents the correlation between the cardiolipin synthesis rate and growth rate. The reduced model of SEFBA is suitable to describe the changes of cell length with growth rate under various growth stresses. The parameter C in Equation (Equation 31) depends on *f*_PG_ and *K*_M_. As both *f*_PG_ and *K*_M_ are about constant, the value of C is 2.96 under different growth stresses. (**A**) The lines display the relationship between growth rate and cell length under different stress conditions. Different types of lines were drawn with different parameters written in the table beside. (**B**) Comparison of the reduced model and experiments. The experimental data were cited from Basan et at. [10].

**Figure 4 cells-09-00391-f004:**
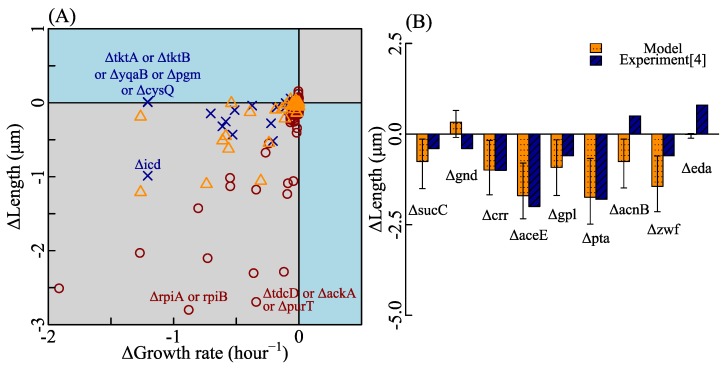
Effects of metabolic gene deletions on growth rate and cell size. (**A**) Each point stands for an *Escherichia coli* mutant with the same parameters in SEFBA. β is equal to 0.22, which is adapted to metabolic networks iJO1366. Other parameters except for β are the same as Figure 2A. Three colors of points stand for three culture media (written in Appendix A). One culture medium corresponds to MOPS medium [6,78] with glucose. One corresponds to M9 medium [25,73,74] with glucose. Another corresponds to rich medium [6] with kinds of amino acids and glucose. The maximum exchange reaction rate of oxygen is 15 mmol/gDW·h, and the maximum exchange reaction rate of glucose is 15 mmol/gDW·h. Detail information of bounds was written in Appendix A. Each mutant can have more than one gene deletion and each gene deletion can also switch off more than one reaction in metabolic reactions network. The detailed information about mutants was written in Appendix A. (**B**) Comparison between experimental data [4] and SEFBA results of nine mutants of metabolic gene deletions. Twenty kinds of bounds were used in SEFBA to simulate different culture media (Appendix A). For each kind of bound, we varied the glucose exchange rate to have the predicted growth rate of FBA close to the experimental data. The ordinate is the variation in cell length of *Escherichia coli strains* with gene deletions compared to that without gene deletions. The error bars are the minimum and maximum values of cell length variation among 20 kinds of culture media.

**Table 1 cells-09-00391-t001:** Parameters and variables involved in this work.

Parameters	Description	Values
*r* _CLPNS_	the cardiolipin biosynthesis rate	from FBA (Section 2.3.1) or as a variable (Section 2.3.2)
*K* _M_	parameter of Michaelis–Menten kinetics	$2\times 10^{-35}\;{\left(\mathrm{mol}/\mu m^{2}\right)}^{2}$ [62] *
*V* _max_	parameter of Michaelis–Menten kinetics	0.38 mmol/(gDW·h) for BW25113 [62,69] *;
		0.3 mmol/(gDW·h) for MG1655 [62,69] *;
		0.76 mmol/(gDW·h) for SJ202;
		0.38 mmol/(gDW·h) for NCM3722
*s_p_*	average surface area covered by a phospholipid molecule	0.5 nm^2^ [54] *
*f_l_*	the fraction of surface area covered by phospholipid in membranes	from FBA based on Equation (Equation 25)
β	coefficient in Equation (Equation 24)	0.5 for iJR904 and 0.22 for iJO1366 based on the experimental data of *f_l_* [54,72] *
*f* _PG_	ratio of the amount of PG to the amount of all kinds of phospholipid located in the cell side surface	18% [58] *
*n_0_*	average amount of phospholipid in cell caps during a cell cycle	from cell width based on Equation (Equation 5)
ϵ	average number of cell caps during a cell cycle	3.4 (fitting)
*D*	cell width	fitting the experimental data of cell width [6,9,13] or as a variable
*C*	$C=\frac{s_{p}\cdot N_{A}}{4\cdot f_{\mathrm{PG}}\cdot f_{l}}\cdot K_{M}^{\frac{1}{2}}$	2.96 *
*H*	$H=1-\frac{v_{0}}{V_{\max}}$	1 based on FBA (Section 2.3.1) or as a variable (Section 2.3.2)
*E*	$E=\frac{w_{\mathrm{CLPNS}}}{V_{\max}}$	fitting

* Obtained from the literature.

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
