# Peer review of "Quantitative Connection between Cell Size and Growth Rate by Phospholipid Metabolism"

_cells, 2020, doi:10.3390/cells9020391_

Round 1

Reviewer 1 Report

The only point previously questioned has been properly addressed. The manuscript is ready for publication

Reviewer 2 Report

No further comments, since I agreed with the previous version.

This manuscript is a resubmission of an earlier submission. The following is a list of the peer review reports and author responses from that submission.

Round 1

Reviewer 1 Report

This is an interesting manuscript describing an FBA-based appoarch to predict cell length/size and its correlation to cell growth rate in E.coli. The approach has potential but certain clarifications and improvements are necessary before the present manuscript can be published. 

On page 2 the authors stipulate that the ratio of membrane constituents is proportionate to the ratio of active transport to diffusion rates. However, the rate of diffusion also depends on the concentration gradient across the membrane of the particular analyte being transported. The authors should provide some evidence in support of this assumption to demonstrate its validity.

On page 7 the authors state that 'The bounds of exchange reactions in FBA correspond to the availability154 of nutrients in growth media'. how were the uptake rates calculated? did the authors assume that all nutrients in the media that were included in the FBA were exhausted and, if so, at what rate - the choice of time interval for these calculations would have tremendous impact on the constraints used to solve the FBA problem. Also, not all nutrients are typically exhausted. The authors themselves recognise that 'Some metabolic reaction rate of SEFBA simulations with high growth rates may be too large to satisfy171 thermodynamic constraint'. Are there no spent media analyses in the literature for E.coli? Given that most FBA approaches are initially developed and tested on E.coli as a model organism I am sure there is a multitude of publications the authors can obtain such uptake rates from.

In the discussion of the reslts presented in Figure 4, the authors state that 'In most cases, the experimental data fall within the values of cell length calculated by SEFBA with247 different media'. however, the error bars are too large to make any solid conclusions, while in certain cases the directionality of the effect is incorrect. The authors need to identify the shortcomings of their method and discuss what may be contributing to these incorrect predictions. The use of calculated instead of measured uptake rates for one is a major flaw of this study because the FMA model is not appropriately constrained. Even though an attempt to mitigates this by analysing different media formulations was made, this again assumes nutrient exhaustion within an undefined time interval. Aklthough I am not an E.coli expert, I believe that the Sauer lab at ETH have generated comprehensive datasets for nutrient utilisation rates, which may be of value to this study.

Minor comments

English language should be checked and improved throughout manuscript.

On page 3, line 76, can the authors please explain what they mean by 'cell side surface' and 'side surface area'?

Figure 3 legend: define D H and E in tables

Section 3.5 contains no references to the experimental studies for mutants apart from Westfall and Levin. is that the only one considered?

Figure 4: include reference for experimental paper in legend.

Reviewer 2 Report

The manuscript by Zhang et al reported a model that describes a relationship between growth rate and cell size via lipid biosynthesis. The claim of this study “Cell size correlated to growth by metabolic network” was not supported by this manuscript following reasons.

Tautology. The rate for the cardiolipin biosynthesis (rCLPNS) was linearly correlated with the growth rate due to a rule of FBA (fig. 1E). As shown in Eqn 17, the cell size L was correlating with rCLPNS. It means that cell size correlated with growth rate because the growth rate correlated with cell size. Arbitrary selection of cardiolipin biosynthesis. There is no biological reason to focus on the cardiolipin biosynthesis instead of other membrane lipids such as PG. Since PG is also important membrane lipid, a similar formulation could be done by the metabolic reaction in the PG biosynthetic pathway. The cardiolipin biosynthesis seems to be selected arbitrary, to introduce a nonlinear relationship in equation 7 that has no biological justification. What is wCLPNS? By introducing Eqn 18, and allowing negative values for wCLPNS, a negative correlation between growth rate and rate for the cardiolipin biosynthesis could be introduced without a biological basis. Fig. 3 means that cell size and growth rates were negatively correlated because parameter E (=wCLPNS) was set to a negative value.

Reviewer 3 Report

As an overall comment, the whole work may reveal to be quite hard to read for the ones not coming from the biological field. Indeed, what is lacking is a cartoon of the cell physiology, clearly explaining the different roles of $s_{area}$, $s_{lipid}$, $s_{protein}$ and so on: Figure 1A completely lacks of clarity. Another important point is that it must be collected in a clear Table how the model parameters are taken inside the main formula, that should be eq.(10). Some are taken from the literature, some other are taken from the Flux Balance Analysis, some come from the manuscript computation, etc. The reader must be able to clearly understand what is the main formula, and how to compute the cell length according to the available literature and computations. Finally, the manuscript could be hard to read even for the ones skilled in mathematical computational tools. Even if computations have been carefully checked and there seem to be no fatal errors, not all the passages seem properly addressed (details and remarks are given to the Comments to Authors). In summary, according to this reviewer, the manuscript cannot be accepted in the present form. The Authors are suggested to carefully address the points highlighted in the review before to resubmit the manuscript.

Remarks.
1) The manuscript is full of mistakes and grammar errors. Just to cite a bunch: 'fond' instead of 'found', 'cover' instead of 'covers', 'exist' instead of 'exists', 'conficients' instead of 'coefficients', 'discribe' instead of 'describe', 'constriant' instead of 'constant'

2) Eq.(4): it is not clear where does 1/4 comes from

3) Row 74, page 3. The measurement unit for $K_M$ is the square of the measurement unit for $[PG]$, therefore it should be $(mol/surface)^2$; instead, it is written as $(mol/length)^2$.

4) Page 4, below eq.(9). Why $s_{area}=\pi DL$? In a flat ellipsoid the surface is approximated by $2\pi DL$. Where does your formula is taken from? What is the geometric shape assumed for E. coli? A prolate ellipsoid? A flat ellipsoid? Even more obscure is the formula for $s_{side}$. A figure (or a cartoon) is here required

5) FBA is invoked to address computation. To this aim, the Authors require a model for the fluxes. Actually they consider two models taken from the literature. Why did they take 2 models instead of just one? Are their results consistent with both models? What if one considers a different model? IN other words: is their approach independent of the choice of the metabolic model?

6) At the end of page, the AUthors do not summarize, just repeat the same stuff previously introduced

7) Page 5: what is the meaning of the sentence 'Neglecting the impact of $n_0$ and $f_1$ on cell size'? Is it a working hypothesis (i.e. an approximation introduced by the Authors) or is it motivated by the literature? Is it reasonable?

8) Page 5: what is the meaning of the sentence 'Theoretically, $w_{CLPNS}$ can be negative'? Is it a mathematical possibility, or is it a biological possibility?

9) Page 6. Figure 1B is misleading. The idea is good, but it has not been accurately addressed. For instance, Why do you use different notation from the manuscript? $N(PG)$ instead of $n_{PG}$. What is $S$? Is it $S_{area}$? What is $r$?

10) What are MOPS? It seems they have not been defined before

11) Reaction (A2): where does the arrow point to?

12) Appendix A.3. The use of $r$ and $r'$ is misleading. It seems that $r$ and $r'$ differ only for the measurement units (see page 19). Is it true? In any case, it is not clearly explained, and it creates confusion. Please rewrite the whole story without ambiguity.

13) Eq.(A35) seems wrong and useless

14) Eq.(A42): at the end, it is not ${s_p\cdot N_A\over 6}$, instead ${s_p\cdot N_A\over 2\epsilon}$

Reviewer 4 Report

Overall, the topic is interesting for the reader but there are some issues that need improvement. Moreover, English language needs to be improved also. For the limited time I had for the review, I recommend "major revision".

(1) There are assumptions that need to be justified: in Eq (3) the ratio of two rates is assumed to be proportional to the ratio of two areas of the cellular system.
(2) Eq (2): Where comes factor 1/4 from?
(3) Notation of variables is confusing: what is the difference between s_area and s_side?
(4) In Fig 1 (C) a simulation is shown with glucose concentration as input, however, the
model equation is missing. The unit given for glucose is wrong, because
it should be a concentration (e.g. mmol/l).

Round 2

Reviewer 1 Report

Author Response 1: It’s an important question. Only small molecules, such as oxygen, water and carbon dioxide, move across the membrane by simple diffusion process

Glucose can also be transported through diffusion and active transport. I understand that this assumption has been removed, but it's not clear to me what the implications are in terms of change in the values of the results.

Author Response 2: The uptake rates calculated by maximizing growth rate and doing flux balance analysis. Bounds of nutrient uptake reactions are the input of FBA. It was set similar to the previous studies of FBA [25, 71, 72, 74]. All the bounds used in this work are written in the Supplementary Materials S1 Table 1,2,4.

I fully understand how FBA works but the values in the Supplementary material for nutrient uptake rates are 0, 1 or 100. How were these calculated? How did you use the experimental data from references 25, 71, 73 and 74? It is not at all clear to me how you revised your calculations. 

Reviewer 2 Report

The responses by the authors failed to address my comments. Thus it is difficult to do further review.

Reviewer 3 Report

Most comments and remark have been properly addressed. This reviewer finds only one point still crucial for the manuscript to be accepted, and it involves Fig.1A, which is still lacking what this reviewer asked, that is a cartoon explaining the relationships among $s_{area}$, $s_{lipid}$, $s_{protein}$. The new Fig.1A is of help for some other points (e.g. to understand the shape of the cell) but it still lacks its primary role to describe the relationships among $s_{area}$, $s_{lipid}$, $s_{protein}$ is a pictorial way

Reviewer 4 Report

My points of criticism are addressed by the authors. So, I have no further comments.